# Preparation, Characterization, and Performance of Natural Zeolites as Alternative Materials for Beer Filtration

**DOI:** 10.3390/ma16051914

**Published:** 2023-02-25

**Authors:** Oana Cadar, Irina Vagner, Ion Miu, Daniela Scurtu, Marin Senila

**Affiliations:** 1INCDO-INOE 2000, Research Institute for Analytical Instrumentation, 67 Donath Street, 400293 Cluj-Napoca, Romania; 2National Research and Development Institute for Cryogenic and Isotopic Technologies Romania, 4 Uzinei Street, 240050 Ramnicu Valcea, Romania; 3SC Utchim S.R.L., 12 Buda Street, 240127 Ramnicu Valcea, Romania

**Keywords:** clinoptilolite, beer filtration, filter aid, clinoptilolite, diatomaceous earth, metals in beer, kieselguhr

## Abstract

The clarity of the beer is essential to its marketability and good consumer approval. Moreover, the beer filtration aims to remove the unwanted constituents that cause beer haze formation. Natural zeolite, an inexpensive and widespread material, was tested as a substitute filter media for diatomaceous earth in removing the haze constituents in beer. The zeolitic tuff samples were collected from two quarries in Northern Romania: Chilioara, in which the zeolitic tuff has a clinoptilolite content of about 65%, and the Valea Pomilor quarry, containing zeolitic tuff with a clinoptilolite content of about 40%. Two-grain sizes, <40 and <100 µm, from each quarry were prepared and thermally treated at 450 °C in order to improve their adsorption properties and remove organic compounds and for physico-chemical characterization. The prepared zeolites were used for beer filtration in different mixtures with commercial filter aids (DIF BO and CBL3) in laboratory-scale experiments, and the filtered beer was characterized in terms of pH, turbidity, color, taste, flavor, and concentrations of the major and trace elements. The results showed that the taste, flavor, and pH of the filtered beer were generally not affected by filtration, while turbidity and color decreased with an increase in the zeolite content used in the filtration. The concentrations of Na and Mg in the beer were not significantly altered by filtration; Ca and K slowly increased, while Cd and Co were below the limits of quantification. Our results show that natural zeolites are promising aids for beer filtration and can be readily substituted for diatomaceous earth without significant changes in brewery industry process equipment and protocols for preparation.

## 1. Introduction

Beer is one of the oldest known beverages and the most consumed alcoholic beverage worldwide [1,2]. Beer comprises a variety of compounds that offer both appreciated sensory features and health benefits. Recently, interest in beer has increased because it is rich in nutrients (carbohydrates, amino acids, minerals, vitamins, and polyphenols) due to the multistep brewing and fermentation process [3]. There are several stages in the process of beer brewing to transform a starchy material into a sugar-rich malt and then, by yeast fermentation, into an alcoholic liquid, the most important being wort production, alcoholic fermentation, maturation, filtration and/or stabilization [2,4]. Over the years, the brewing process has experienced novel technological advances, and consumer preferences have permitted the manufacture of a wide variety of beers more effectively [5].

Except for the unfiltered beer class (craft beers), high turbidity in beer (lagers and pilsners) is regarded as an inferior quality, with beer clarity being a basic precondition of its good marketability and consumer satisfaction [6,7]. Accordingly, this operation has a high accent in breweries and minibreweries [2,7,8]. The filtration step plays an important role during the beer production by eliminating unsought constituents from the beer in order to obtain a product of the highest value. In order to be considered clear and brilliant, beer should form no haze when chilled to 4 °C or below [9]. When considering their dimension, haze particles are divided into the macroscopic visible haze, with a particle size > 1 µm (yeasts, microorganisms, etc.), colloidal particles, with particle size < 1 µm (protein, tannins, gums, etc.), and microscopic particles, with particle sizes < 0.001 µm [10].

Next to other characteristics such as taste, color, flavor, or foam, the clarity of filtered beers is a very important quality parameter [11,12]. The filtration process should eliminate the suspended materials or haze-active substances (proteins, tannins, yeasts, etc.) that cause turbidity, yet the compounds that give the characteristic beer taste, color, flavor, and foam should not be altered [4]. In addition, filtration ensures beer stability, which is almost impossible to be reached in the order of months in unfiltered beers, even if these are stored in ideal conditions (temperatures up to 5 °C and no sunlight) [7].

At the industrial level, beer is passed through filters in order to eliminate the suspended particles and disturb the potential turbidity producers (stabilization) [2]. Most breweries depend on precoat filtration using diatomaceous earth or diatomaceous earth as the filter aid [7,10,13,14,15]. While diatomaceous earth offers many advantages (chemical composition, particle size, dry and wet densities, water absorption capacity, pH value, and color), it also has disadvantages, namely disposal, health and safety concerns, and limited resources [10,16]. Therefore, there is a need for alternative materials with similar filtering properties of diatomaceous earth in order to be used as a filter media in the existing brewery industry process equipment and protocols with minimal adaptation.

Natural zeolites appear to be a suitable substitute filter media for diatomaceous earth in removing haze constituents in beer [10]. Natural zeolites are low-cost and large reserves and are microporous materials composed of a three-dimensional framework constructed of SiO_4_ and AlO_4_ tetrahedra, which have many valuable features, such as high porosity and bulk volume, a porous and permeable structure, chemical resistance, high purity, large specific surface area, high adsorption capacity, and good adsorptive properties for use in industrial, environmental, agricultural, or medical applications [17,18,19]. Shafrai et al. reported that 4.5% zeolite (as a stimulating additive) guarantees the maximum accumulation of cells with glycogen and budding yeast cells and the minimum content of dead cells in beer wort when stored for no more than two days [6]. Remarkably, natural zeolites can be readily substituted for diatomaceous earth without significant changes in brewery industry process equipment and protocols for preparation [2,13,20]. Unlike diatomaceous earth, which is classified as a severe carcinogen with its attendant health risks, natural zeolites are nontoxic and safe materials for the environment when disposed of [21]. The scientific literature regarding the use of zeolites as filtering materials for beer is very scarce. In a previous study, two size grades of synthetic zeolite-A (125–250 µm and 63–125 µm) were experimentally assessed against diatomaceous earth to clarify beer [20]. Hydrophobic ZSM-5 type zeolites were used as molecular sieves to remove (by adsorption) the volatile flavor compounds (i.e., aldehydes) that affect the sensorial quality of beer [22,23]. Permyakova et al. [24] used natural zeolite to adjust wort composition. They reported an acceleration in the fermentation process and an improvement in the final product quality due to the removal of phenols and high-molecular-weight proteins from the wort by the zeolites. However, to the best of our knowledge, no literature data have reported using natural zeolites as filtering media for beer.

Natural zeolites have different physicochemical properties according to their source and, in addition, may contain traces of some potentially toxic metallic ions [25]. Thus, using of natural zeolites in the food industry involves strict quality control. Considering all these aspects, this study presents, for the first time, the use of natural zeolites as substitutes for diatomaceous earth in the beer filtration process. In order to evaluate the possible influence of the provenience of zeolitic tuffs on the filtration capabilities, natural zeolites from two different deposits from Northwest Romania were tested as beer filtration aids after improving their features. The physicochemical properties of thermally treated zeolites were determined. The filtration characteristics of the natural zeolites and their mixtures with commercial filter aids were investigated in laboratory experiments using unfiltered beer samples. The results were then compared with those of the commercial filter aids available on the market.

## 2. Materials and Methods

### 2.1. Chemicals

All reagents used were of analytical grade. 37% (m/m) HCl, 65% HNO_3_ (m/m), and 40% (m/m) HF from Merck, Darmstadt, Germany, were used for sample digestion. For metal determination, the calibration standards of 0–20 mg L^−1^ were prepared from multielement standard ICP solutions of 1000 mg L^−1^ (Merck, Darmstadt, Germany) by appropriate dilutions. Ultrapure water from a Purelab flex 3 system (Buckinghamshire, UK) was used to dilute the samples and to prepare the calibration solutions. The accuracy in determining the total element concentrations in zeolite was evaluated via BCS-CRM 375/1 soda feldspar (Bureau of Analyzed Samples, Middlesbrough, UK), achieving satisfactory recoveries (%) of all analytes (89.6–102.6%).

### 2.2. Zeolite Collection, Preparation, and Characterization

The zeolitic tuff samples were collected from Chilioara (CS) (Figure 1a) and Valea Pomilor (VP) (Figure 2a) and quarries located in Salaj County, Northern Romania. The rocks were crushed and sieved using a vibratory disc mill RS 200 (Retsch, Haan, Germany) to obtain a particle size < 1 mm and dried at 105 °C. Subsequently, the samples were micronized at a pressure of 12 bar using a PilotMill-2 system (Como, Italy) to obtain fine-size (<40 µm) and rough-size particles (<100 µm) from each quarry. The zeolite samples were named CS40 (Figure 1b), CS-100 (Figure 1c), VP40 (Figure 2b), and VP100 (Figure 2c), according to their provenience and size grades. The samples were thermally treated at 450 °C for 4 h in the air for activation in order to improve their adsorption property and removal of organic compounds.

The concentrations of major (Al, Fe, Na, K, Ca, and Mg) and trace (Cd, Pb, Zn, Ni, Cr, and Cu) elements were determined using an Optima 5300 DV (Perkin-Elmer, Woodbridge, ON, Canada) inductively coupled plasma-optical emission spectrometer (ICP-OES), next microwave-assisted digestion with a mixture of HNO_3_ 65%:HCl 37%:HF 40% (3:9:2, *v*:*v*:*v*) [26]. The concentrations of major elements were converted to oxides by multiplying with 1.8895 (Al_2_O_3_), 1.4297 (Fe_2_O_3_), 1.3392 (CaO), 1.6583 (MgO), 1.2046 (K_2_O), 1.3480 (Na_2_O), and 1.2912 (MnO), considering the atomic and molecular masses. The SiO_2_ and loss of ignition (LOI) were determined by the gravimetric method [19]. The cation exchange capacity (CEC) was calculated from the measured concentrations of the major extractable cations (K^+^, Na^+^, Ca^2+^, and Mg^2+^) using ICP-OES, after their extraction in ammonium acetate solution 1 M [27,28]. Three parallel measurements (*n* = 3) were carried out for each sample/parameter. Total surface area, total pore volume, and pore radius were obtained from N_2_ adsorption–desorption isotherms using a Sorptomatic 1990 apparatus (Thermo Fisher Scientific, Waltham, MA, USA) [29]. The X-ray diffraction (XRD) patterns were recorded using a D8 Advance (Bruker, Karlsruhe, Germany) diffractometer with CuKα radiation (λ = 1.54060 Å), operating at 40 kV and 40 mA.

### 2.3. Laboratory-Scale Beer Filtration Experiments

Commercially unfiltered beer was used for the filtration experiments. The beer was stored in a refrigeration unit at 6 ± 2 °C and was degassed on a magnetic stir prior to the filtration experiments. The filtration mixtures were prepared using zeolitic tuffs and commercially available filter aids, namely DIF BO (rough-size kieselguhr D90 ≤ 220 µm) and CBL3 (fine-size kieselguhr, D90 ≤ 60 µm) from CLARCEL (Chemviron, Feluy, Belgium). The mixtures of filter aids used in 21 different filtration experiments are presented in Table 1.

A lab-scale filtration system (Figure 3) consisting of glass-fritted filter support for the filter (12.5 cm^2^ filtration area) and a funnel with a nominal volume of 250 mL connected to a manual vacuum pump at a pressure of 0.2 bar (Sartorius, Göttingen, Germany) was used.

To simulate the filtration conditions of beer in industry, a mineral filtration layer (filter cake) was deposited on a filter support, and doses of fine-size material (filtration dosage) mixed with the beer before filtration (to ensure beer clarification) were used. Filter papers (wide pore size) were used on the filter support to maintain the filtration layers. 3 g of filtering materials in different mixture ratios (Table 1) were used to form the filtration layer. Firstly, 50 mL of ultrapure water was mixed with the filtering material and added to the filter in order to remove the very fine particles for the filtering material and form a uniform layer of filter cake. 0.5 g of fine-size filtering aids (kieselguhr or zeolite) were used as filtration dosage to form a slurry with 100 mL of beer (approximately 0.5% by weight) before filtration. Filtered beer was collected until wholly liquid was filtered, and the filtration time was recorded (Table 1). The experiments were carried out in triplicate.

### 2.4. Filtered Beer Characterization

Filtered and unfiltered beers were characterized in terms of turbidity (EBC), color (EBC), pH, the concentration of major elements (Ca, Mg, Na, K), and trace elements (Fe, As, Cd, Co, Cr, Cu, Ni, Pb, Zn). The turbidity (permanent haze, Hp) was measured using a turbidity meter (Turb 555 IR, Weilheim, Germany) following the European Brewery Convention (EBC) method 9.29 (1 EBC = 0.25 NTU) [30]. Turbidity was determined at a wavelength of 470 nm and expressed in nephelometric turbidity units (NTU). Four formazine-based calibration standards having turbidity of 0, 8, 80, and 800 NTU were used. The color was calculated in EBC units after the measurement of beer absorbance at 430 nm using a Lambda 25 (Perkin Elmer, Norwalk, CT, USA) spectrophotometer [31]. Beer pH was determined by a pH meter (Mettler Toledo, Schwerzenbach, Switzerland). Sensory analysis was performed by a triangle test in accordance with the EBC method no. 13.7 (Triangle Test) [30]. The concentration of major and trace elements in beer was measured after wet acid digestion (HNO_3_ 65%:H_2_O_2_ 30%, 3:1, *v*:*v*) using a closed-vessel Speedwave Xpert microwave system (Berghof, Eningen, Germany). The concentrations of Na, K, Ca, Mg, and Fe were measured by ICP-OES (Optima 5300 DV Perkin Elmer, Woodbridge, ON, Canada) using calibration standards in the range 0–20 mg L^−1^ prepared from a 1000 mg L^−1^ multielement ICP solution (Merck, Darmstadt, Germany) by appropriate dilutions. The concentrations of As, Cd, Co, Cr, Cu, Ni, Pb, and Zn were measured by GFAAS (Perkin Elmer model PinAAcle 900T (Norwalk, CT, USA)) [32] using calibration standards prepared from 1000 mg L^−1^ mono element standard solutions (Merck, Darmstadt, Germany) by appropriate dilutions. All analytes exhibited highly linear responses (R^2^ > 0.9990). To check the possible influence of the sample matrix on the analytes signals, 1 mg L^−1^ of each analyte was added to a solution measured by ICP-OES. Recoveries were 95% for Na, 91% for K, 106% for Ca, 93% for Mg, and 98% for Fe, respectively. To check the matrix effects in atomic absorption spectrometry, 1 µg L^−1^ of each analyte was added to a solution measured by GFAAS. Satisfactory recoveries were obtained, namely 91% for As, 98% for Cd, 88% for Co, 93% for Cr, 103% for Cu, 104% for Ni, 96% for Pb, and 108% for Zn. Precision for instrumental determination as determined by % CV for both techniques was ≤5%.

## 3. Results and Discussion

### 3.1. Zeolite Physicochemical Characteristics

The main physical properties of the micronized and thermally treated zeolitic tuff samples in terms of the total surface area, total pore volume, and average pore radius are presented in Table 2.

The total surface area determined by the BET method in the Chilioara samples was 73 m^2^ g^−1^ in CS100 and 72 m^2^ g^−1^ in CS40, while the total pore volume was 0.187 cm^3^ g^−1^ in CS100, and 0.160 cm^3^ g^−1^ in CS40, respectively. The average pore radius was 27 Å in both the CS100 and CS40 samples. In the samples from Valea Pomilor, the total surface area was smaller by about 25% than the samples from Chilioara: 55 m^2^ g^−1^ in VP100 and 53 m^2^ g^−1^ in VP40. Additionally, the total pore volume was approximately 30% smaller than the Chilioara samples, namely 0.119 cm^3^ g^−1^ in VP100 and 0.114 cm^3^ g^−1^ in VP40. The average pore radius was 31 Å in both the VP40 and VP100 samples. Since the pore radius is less than 50 nm and in accordance with the classification established by the International Union of Pure and Applied Chemistry (IUPAC) [33], all samples have mesoporous structures.

The total surface area was slightly reduced in the samples with grain sizes < 40 µm compared to the samples with grain sizes < 100 µm; a possible explanation could be that the surface area of porous materials is divided into the external and internal specific surface areas and even thought by advanced grinding the external surface area growths, the internal surface area which has the most significant contribution to the total surface area remains almost unchanged [33,34]. The surface area and total pore volume are linked with the adsorption process and offer spaces for retention [35,36]. Therefore, according to the obtained results, the zeolitic samples from Chilioara should offer more places to retain different substances from the filtered liquids.

The XRD patterns of the zeolitic tuffs show the presence of K-clinoptilolite (PDF 01-080-1557) as the main phase, attended by muscovite (PDF 00-060-1516), albite (PDF 00-020-0548), orthoclase (PDF 00-031-0966), quartz (PDF 01-070-7344), and montmorillonite (PDF 00-058-0548) (Figure 4). The XRD pattern displays the characteristic peaks of the clinoptilolite zeolite structure (2θ at around 10, 25, 26, 30, and 32°) [19]. The RIR (reference intensity ratio) method was used for the quantitative phase analysis, indicating a clinoptilolite content of 60–65% for CS and 35–40% for the VP samples, respectively. The degree of crystallinity is similar at <100 and <40 µm in both cases but significantly higher for CS (approximately 65%) when compared to VP (approximately 5%) samples. The occurrence of amorphous volcanic glass in the zeolitic tuffs is indicated by the broad hump centered near 25° (2θ) [37].

The chemical composition, in terms of the major oxides and inorganic components, pH, exchangeable cations, and total exchange capacity of the four samples (CS100, CS40, VP100, VP40) thermally treated at 450 °C, are presented in Table 3.

According to the SiO_2_/Al_2_O_3_ ratio, the zeolitic tuffs from Chilioara and Valea Pomilor are in the group of super silicic materials [38]. The grain sizes do not significantly influence the chemical composition of the filtering materials obtained from the same quarry. The samples from Valea Pomilor generally contain higher amounts of Na and K than those from Chilioara, while the contents of the other major constituents (Si, Al, Ca, and Mg) are comparable. Mostly, the content of the trace and ultra-trace elements (Fe, Mn, Cr, Cu, Co, Ni, Pb, and Zn) are higher in the samples from Valea Pomilor than in those from Chilioara, while the concentrations of As and Cd are below the corresponding limits of quantification in all the samples.

The cation exchange capacity (CEC), calculated as the sum of the exchangeable concentrations of the major cations (K^+^, Na^+^, Ca^2+^, and Mg^2+^) extracted from the 1 M ammonium acetate solution, was higher in the samples from the Chilioara quarry, mainly because of the lower exchangeability of Ca^2+^ in the VP40 and VP100 samples. In all the samples, the main contribution to the total CEC was represented by Ca^2+^, followed by K^+^ (about 18 meq 100 g^−1^), while Na^+^ and Mg^2+^ had minor contributions. Since the total concentrations of Na^+^, K^+^, Ca^2+^, and Mg^2+^ measured by acid digestion were generally higher in the samples from Valea Pomilor, the lower CEC values could be explained by the different mobility of these cations from the two sources. In the samples from Chilioara, the percent of exchangeable Ca^2+^ was in the range of 87–98% of the total content, the percent of exchangeable K^+^ was approximately 35% of the total content, and the percent of exchangeable Na^+^ was about 23%, while about 12–15% of total Mg^2+^ were exchangeable. In the samples from Valea Pomilor, the percent of exchangeable Ca^2+^ was in the range of 63–74% of the total content, the percent of exchangeable Na^+^ was approximately 47%, the percent of exchangeable K^+^ was approximately 36%, and only about 4–6% of Mg^2+^ were exchangeable. These results indicate which cations are exchangeable and can be involved in the exchange processes.

### 3.2. Main Physical-Chemical Characteristics of Filtered Beer from Laboratory Tests

Based on the laboratory tests and beer analysis, the influence of filtering material mixtures on the main parameters of the beer is presented in Table 4. The filtration time generally increases when only zeolites are used to form the filter cakes. Additionally, using smaller particle sizes as filtering materials increases the filtration time. A variation from 16 min (when only rough-size kieselguhr DIF BO was used as the filter cake) to 28 min when mixtures of fine-size particles (<40 µm) and rough-size particles (<100 µm) zeolites were used to form the filter cake was recorded. According to the data for experiments E1–E3, the material used for beer dosage (CBL3, VP40, or CS40) does not significantly influence the filtration time.

The pH of the unfiltered beer was 4.20, and generally, this very slowly increased because of interaction with the filtering material. The highest pH value was measured in the beer filtered in experiment E16, in which the pH increased by 0.26 units, probably due to the release of some alkaline cations (Na^+^, K^+^) from the filtering material to the beer. On average, an increase in beer pH of 0.2 units was observed after filtration, but the beer pH remained in the acceptable range of pH 4.0–4.4 [39].

The taste, specific smell, and color of beer are given by colloidal particles with sizes of 0.5–3 μm [40]. Particles with higher diameters are significantly correlated with nephelometric haze values. Thus, to obtain a beer of satisfactory quality, it is necessary to maintain colloidal particles that provide the specific taste, smell, and color and to separate (by filtration) the larger particles that affect the beer turbidity and its long-term stability. According to the physical properties presented in Table 2, the average pore radius value of the zeolites (27–31 Å) is much smaller than the colloids, which should be maintained in beer, and, consequently, the retention of these particles is avoided, while the large particles from beer (haze) can be retained on the external surface of filtering material.

In all the experiments, the taste and flavor of beer remained mainly unaffected by filtration (Table 4). These parameters were evaluated by triangle testing, a sensory evaluation to blindly identify any differences between samples. Briefly, for each test, within the three samples presented to each assessor, two were similar. Particularly, for the beers filtered using only zeolites, the perception of the wort flavor of the beer was reduced, probably due to the removal of some aldehydes that resulted from beer fermentation. This sensory perception was also previously reported when zeolites were used as molecular sieves to remove the volatile flavor compounds or during wort fermentation [22,23,24]. The beer color generally decreased in intensity from 14.1 EBC (amber) to a gold color (EBC value 8 to 12) in the beers filtered in most of the filtration experiments. The beers filtered in experiments E9, E14, E15, E20, and E21 had a yellow color (EBC value 6 to 8). As a particularity, in those experiments, only zeolites were used to obtain the filter cakes.

The European Brewery Convention recommends a standard haze for diatomaceous-earth-filtrated beer of less than 0.6 EBC units is recommended [30,41]. The turbidity of the unfiltered beer was 14.1 EBC units, which means a very hazy beer (EBC > 8.0). In our experimental conditions, the beer filtered using only DIF BO (rough-size kieselguhr) as the filter cake and CBL3 (fine-size kieselguhr) as the filtration dosage had the higher turbidity (1.51 EBC) in the scale of very slightly hazy (EBC 1.0–2.0). Additionally, the beers filtered in experiments E2, E3, E12, and E13 were slightly hazy (1.25 EBC, 1.12 BC, 1.11 EBC, and 1.05 EBC). In most cases, the filtered beers have turbidity in the range of 0.5 EBC to 1.0 EBC, indicating an almost brilliant beer, while in experiments E15, E20, and E21, the beer was brilliant (EBC < 0.5). By increasing the amount of zeolite in the mixtures used as the filtering material, an improvement in the elimination of beer haze was observed. The lower values for turbidity were obtained in experiments E20 and E21 with values of 0.44 EBC and 0.35 EBC, respectively, in which both the filter cakes and filtration dosage were zeolite.

Even if, in some breweries, the beer started to be filtered through 0.45 µm hollow-fiber modules by the so-called Norit process [42], the filtration in most breweries worldwide still uses conventional powder filters has been the standard industrial practice for more than 100 years. According to the laboratory-scale results, zeolites meet the criteria as a substitute for diatomaceous earth when considering the main characteristics of filtered beers: pH, color, turbidity, taste, and flavor, and the filtration time, which were not highly affected. Figure 5 shows a negative correlation between filtration time and beer color (R^2^ = 0.6217, r = −0.7885). Similarly, a negative correlation between filtration time and beer color was observed (R^2^ = 0.6130, r = −0.7829). Consequently, a strong positive correlation exists between turbidity and the color of filtered beer (Figure 6). Using zeolites in high amounts and fine particle sizes increases the filtration time but also retains more colloids from the beer, having the main effect of reducing beer color and turbidity.

### 3.3. Influence of Filtering Media on Metal Content in Filtered Beer

According to the concentration and type, metals may be essential or toxic to the human body andaffect beer quality [43]. Metals such as Fe, Cu, Cr, Ni, Co, and Zn have various biochemical and physiological functions, being essential elements, but can become toxic in elevated concentrations [44,45,46]. Other elements, such as Pb, As, and Cd, may cause severe problems for human health even at low levels and are considered non-essential elements [47,48]. Pb is a highly toxic metal with a negative impact on nearly every organ in the human body, while Cd has no biological role in the human body, affecting mainly the liver and kidneys [49], as it is a toxic metalloid that is widespread in nature, known as an endocrine disrupter and human carcinogen [50]. Metals can enter into beer from the raw materials used for beer production (cereals, water, and yeasts) [51], but the materials used for filtration can also transfer metals into the filtered beer following their interaction during the filtration process, which is another source of toxic elements entering into beers. The concentrations of major elements and trace elements in beer samples are presented in Table 5.

When considering the chemical composition of the natural zeolites used to produce the filter aids used in the laboratory-scale filtration experiments, one of the aims of this study was to evaluate if these elements are transferred into the beer during filtration.

The average concentrations of major elements decreased in the following order: K > Mg > Ca > Na. When compared to the unfiltered beer, the concentrations of K decreased by about 10% in the first three experiments, in which only kieselguhr was used as the filter cake, and increased in all the other experiments, with the highest concentration (615 mg L^−1^) measured in the sample filtered in experiment E9. The concentrations of Na and Mg were not significantly modified by filtration. The concentration of Na in the beer samples ranged between 10.9–20.5 mg/L, much lower than the maximum admitted limit for drinking water of 200 mg L^−1^ [52]. Na concentrations higher than 200 mg L^−1^ can induce a salty taste in drinks. In the case of Ca, the concentrations in the filtered beers generally increased from 21.0 mg L^−1^ to a maximum concentration of 69.1 mg L^−1^ in sample E9, probably due to the higher exchangeability of Ca^2+^ ions in the zeolites. Increasing Ca and K concentrations following filtration does not affect the beer taste. Moreover, these elements have a beneficial effect on health.

Fe concentration is an important parameter in beer; its deficiency is manifested mainly by anemia. On the other hand, increased concentrations of Fe^3+^ ions contribute to the beer’s metallic and astringent taste. As observed in Table 5, the concentration of Fe was not strongly influenced by the filtration process and remained in the same order of magnitude as that from the unfiltered beer. In a previous study on metal content in beers from a Turkish market, Charehsaz et al. [53] reported Fe concentrations in the range of 392.27 to 1454.14 µg/L, close to the Fe concentrations measured in the present study. Higher Fe concentrations were reported by Kostic et al. [54] for wine samples from Serbia, in the range of 2.93–36.2 mg L^−1^. The concentrations of trace elements in beer samples analyzed by GFAAS are presented in Table 6.

The concentrations of Cd and Co in all the analyzed samples were below the quantification limits (LOQs) of 0.26 µg L^−1^ and 0.65 µg L^−1^, respectively. In drinking water, the maximum admitted level for Cd is 5.0 µg L^−1^ [52]; thus, the Cd concentration in filtered beers is well below this limit. The concentration of As was <0.50 µg L^−1^ in the unfiltered sample and increased to concentrations in the range of 0.68–4.05 µg L^−1^ after filtration. Thus, this concentration can be attributed to the filtering materials (both kieselguhr and zeolites), which are used in the filtering stage. However, this concentration is below the maximum limit for drinking water (10 µg L^−1^) [52]. In a previous study on the transfer of As, Cd, and Pb from diatomaceous earth filter aids, Redan et al. [55] reported a significant increase in inorganic As in filtered beverages (beer and wine) in the range of 11.2–13.7 μg L^−1^ following the contact with the filter aids. At the same time, no modifications were observed for the Pb and Cd concentrations. Donadini et al. [56] surveyed As, Cd, and Pb in beer brands from the Italian market. The reported average content of As was 10.82 ± 5.54 µg L^−1^, which is higher than in our study. The Cd concentration was 0.16 ± 0.15 µg L^−1^, while the Pb average concentration was 1.84 ± 3.24 µg L^−1^.

Cr is an essential element with beneficial effects for human health in low concentrations, but it becomes toxic in increased concentrations [57]. Consequently, the maximum admitted level in drinking water for total Cr is 50 µg L^−1^ [52]. In the analyzed beer samples, higher Cr concentration from 2.80 µg L^−1^ in the unfiltered sample to concentrations in the range of 11.2–26.6 µg L^−1^ in filtered beers was observed. Indeed, in all cases, the Cr content was below the maximum limit. Still, it can be observed that the highest Cr concentration was measured in the beer filtered using only kieselguhr as a filter aid. In contrast, the lowest concentration was found in the beer filtered using the zeolite from Chilioara (CS100 and CS40) for filtration. Thus, it can be concluded that zeolite contributes less to a higher Cr content in filtered beer than commercial filter aids.

Cu is recognized as an essential element with a role in the function of necessary enzymes in the organism, such as cytochrome c-oxidase, which is a critical enzyme in energy metabolism [53]. Like other essential elements, in augmented concentrations, this can become toxic; thus, its maximum limit in drinking water was established at 2.000 µg L^−1^ [52]. The concentration of Cu in the unfiltered beer was 57.9 µg L^−1^ and generally decreased in the range of 16.5–50.9 µg L^−1^ after the filtration. The most critical removal of Cu was observed in beers filtered only with zeolites as a filter aid. Ni also became toxic in elevated concentrations, and its maximum limit in drinking water is 20 µg L^−1^ [52]. In the unfiltered beer sample, the concentration of Ni was 10.9 µg L^−1^, and, as was the case for Cu, the concentrations of Ni decreased following filtration. Additionally, the zeolites were found to be more efficient in removing Ni than the commercial filter aids. The concentrations of Ni in the filtered beer samples ranged between 3.11–10.7 µg L^−1^ after the filtration in all cases below the maximum limit for drinking water.

Zn is an essential element as it is implied to produce hundreds of enzymes in the human body [58]. However, exposure to high concentrations of Zn may cause gastrointestinal symptoms or renal damage [58], but a maximum admitted concentration in drinking water exists. Generally, the concentrations of Zn increased in the beers after the interaction with the filter aids, from 10.9 µg L^−1^ in unfiltered beer to a range of concentrations between 17.7–44.1 µg L^−1^ in the filtered beer samples.

The spent filtration aids after beer clarification represent an important issue. Usually, the spent diatomaceous earth is disposed of via landfilling or is directly applied as a fertilizer in agriculture due to the high content of organic matter [59]. Natural zeolites are already used to improve agricultural soil quality [60]; thus, using the spent zeolites from the beer filtration process is easy to implement. However, it should be noted that applying it to soil can release nitrogenous substances that can leach into the groundwater [59]. Moreover, like diatomaceous earth, natural zeolites are nonrenewable natural minerals with limited resources. At present, only limited amounts of waste materials are reused after removing organic loading by thermal or acid/alkaline treatment [61,62]. When considering environmental safety and resource management and correlated with the high thermal and chemical stability of clinoptilolite-type zeolites, future studies on the regeneration procedures of spent zeolites from beer filtration are of great interest.

## 4. Conclusions

Two-grain sizes (<40 and <100 µm) of zeolitic tuff samples collected from two quarries (Chilioara and Valea Pomilor) located in Northern Romania were thermally treated at 450 °C and were used for laboratory-scale beer filtration using different mixtures with commercial filter aids (DIF BO: rough-size kieselguhr and CBL3: fine-size kieselguhr). Beer taste and flavor were not affected by filtration, while the pH very slowly increased because of the interaction with the filtering material, but this remained in the acceptable range of 4.0–4.4 pH units. When using a higher amount of zeolites, the turbidity decreased, and the beer became whiter, yet the color was within the acceptable quality standard. Given the chemical composition of the natural zeolites, we also evaluated whether major and trace elements were transferred into the beer during filtration. Concerning the major elements, the Na and Mg concentrations did not significantly change after filtration, while the Ca and K concentrations slowly increased, but this did not affect the taste of the beer and could even improve the beer fining performance. Regarding the trace elements, an increasing tendency of As, Cr, and Zn concentrations was observed following the filtration; the Cu, Ni, and Pb concentrations decreased after interacting with the filtering material, while the Cd and Co concentrations were below the limits of quantification. In all cases, the concentrations of the trace elements did not exceed the corresponding maximum admitted levels in drinking water. The obtained results lead us to recommend the inexpensive and widespread clinoptilolite-type natural zeolites, micronized and thermally treated, as a suitable substitute filter media for diatomaceous earth to remove haze constituents in beer. However, for their use as commercial filter aids, additional studies are still necessary to verify their behavior at pilot-scale installations for beer filtration.

## Figures and Tables

**Figure 1 materials-16-01914-f001:**
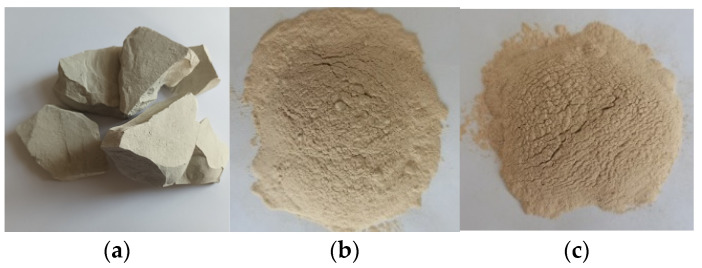
Zeolitic volcanic tuffs from Chilioara quarry before (**a**) and after micronization CS100 (**b**) and CS40 (**c**).

**Figure 2 materials-16-01914-f002:**
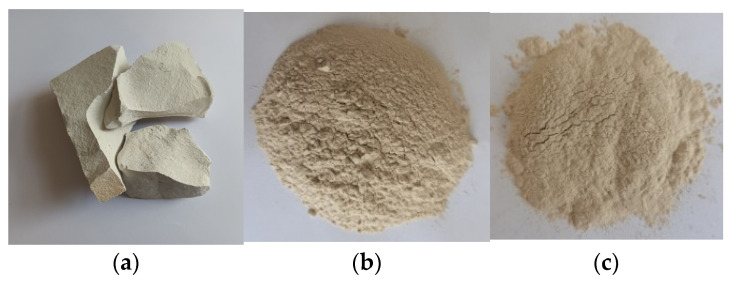
Zeolitic volcanic tuffs from the Valea Pomilor quarry before (**a**) and after micronization VP100 (**b**) and VP40 (**c**).

**Figure 3 materials-16-01914-f003:**
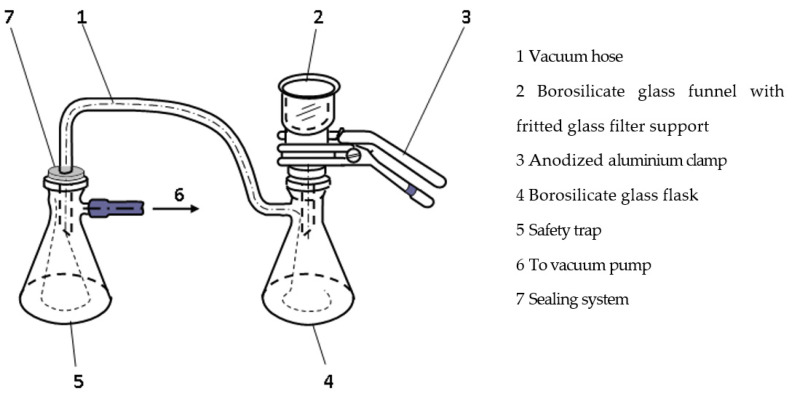
Schematic diagram of the lab-scale filtration system used for beer filtration experiments.

**Figure 4 materials-16-01914-f004:**
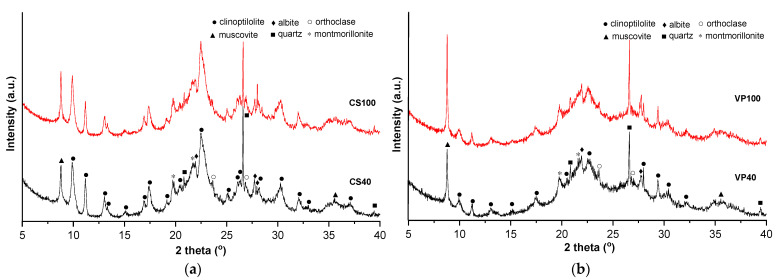
X-ray diffraction patterns of the zeolitic tuff micronized at <100 and <40 µm, thermally treated at 450 °C: (**a**) Chilioara and (**b**) Valea Pomilor.

**Figure 5 materials-16-01914-f005:**
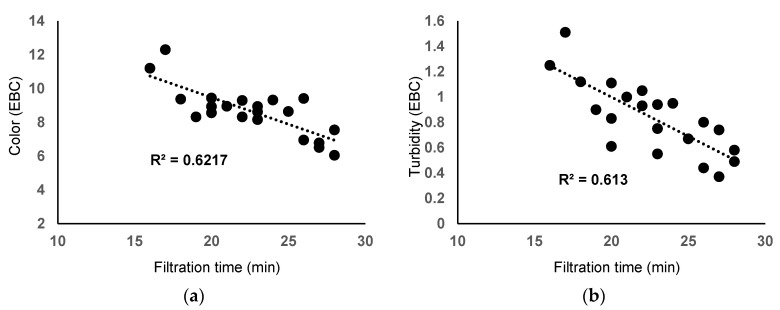
Correlation between filtration time and beer color (**a**) and beer turbidity (**b**).

**Figure 6 materials-16-01914-f006:**
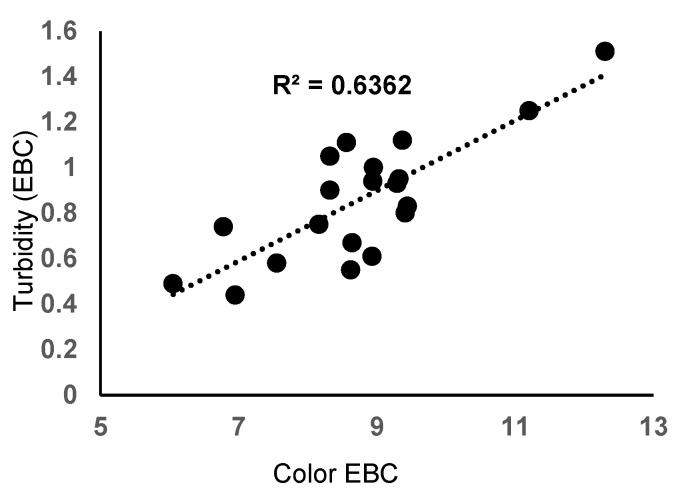
Correlation between beer color and beer turbidity.

**Table 1 materials-16-01914-t001:** Filter aids and quantities used as filter cake and filtration dosage for clarification of 100 mL beer in the filtration experiments.

Experiment	Filter Cake	Filtration Dosage
E1	3 g DIF BO	0.5 g CBL3
E2	3 g DIF BO	0.5 g VP40
E3	3 g DIF BO	0.5 g CS40
E4	2 g DIF BO + 1 g VP100	0.5 g CBL3
E5	2 g DIF BO + 1 g VP40	0.5 g CBL3
E6	2 g DIF BO + 1 g CS100	0.5 g CBL3
E7	2 g DIF BO + 1 g CS40	0.5 g CBL3
E8	2 g VP100 + 1 g VP40	0.5 g CBL3
E9	2 g CS100 + 1 g CS40	0.5 g CBL3
E10	2 g DIF BO + 1 g VP100	0.5 g VP40
E11	2 g DIF BO + 1 g VP40	0.5 g VP40
E12	2 g DIF BO + 1 g CS100	0.5 g VP40
E13	2 g DIF BO + 1 g CS40	0.5 g VP40
E14	2 g VP100 + 1 g VP40	0.5 g VP40
E15	2 g CS100 + 1 g CS40	0.5 g VP40
E16	2 g DIF BO + 1 g VP100	0.5 g CS40
E17	2 g DIF BO + 1 g VP40	0.5 g CS40
E18	2 g DIF BO + 1 g CS100	0.5 g CS40
E19	2 g DIF BO + 1 g CS40	0.5 g CS40
E20	2 g VP100 + 1 g VP40	0.5 g CS40
E21	2 g CS100 + 1 g CS40	0.5 g CS40

**Table 2 materials-16-01914-t002:** Physical properties of the micronized and thermally treated zeolites.

Parameter	CS100	CS40	VP100	VP40
Total surface area (m^2^ g^−1^)	73	72	55	53
Total pore volume (cm^3^ g^−1^)	0.187	0.160	0.119	0.114
Average pore radius (Å)	27	27	31	31
Bulk density (g cm^−3^)	0.726	0.617	0.680	0.598

**Table 3 materials-16-01914-t003:** Chemical properties of micronized and thermally treated zeolites (average ± t × s, *n* = 3 parallel measurements).

Parameter	Measurement Units	CS100	CS40	VP100	VP40
pH	pH	8.04 ± 0.15	8.09 ± 0.17	8.31 ± 0.20	8.42 ± 0.18
SiO_2_	wt.%	68.4 ± 1.7	68.3 ± 2.5	68.0 ± 1.8	67.5 ± 2.1
Al_2_O_3_		10.5 ± 0.8	10.3 ± 0.7	11.6 ± 0.9	12.4 ± 0.8
Fe_2_O_3_		1.22 ± 0.22	1.30 ± 0.15	1.80 ± 0.21	1.66 ± 0.19
CaO		1.62 ± 0.25	1.95 ± 0.23	2.33 ± 0.31	2.21 ± 0.32
MgO		0.79 ± 0.08	0.65 ± 0.07	0.92 ± 0.07	0.74 ± 0.09
K_2_O		2.36 ± 0.43	2.41 ± 0.35	3.66 ± 0.31	3.52 ± 0.40
Na_2_O		0.25 ± 0.03	0.28 ± 0.04	1.14 ± 0.09	1.09 ± 0.10
MnO		0.04 ± 0.01	0.04 ± 0.01	0.07 ± 0.01	0.06 ± 0.01
Na	mg kg^−1^	1890 ± 340	2045 ± 236	8458 ± 987	8087 ± 926
Mg		3910 ± 603	4740 ± 559	5520 ± 734	4440 ± 643
Ca		11,600 ± 1175	13,800 ± 1486	16,640 ± 1266	15,790 ± 1920
K		19,600 ± 3571	19,960 ± 2899	30,370 ± 2572	29,210 ± 3319
Fe		8560 ± 1027	9110 ± 1301	12,600 ± 995	11,620 ± 1066
Mn		331 ± 63	282 ± 51	542 ± 79	465 ± 80
As		<0.50	<0.50	<0.50	<0.50
Cd		<0.10	<0.10	<0.10	<0.10
Cr		3.30 ± 0.31	3.18 ± 0.30	5.44 ± 0.43	6.20 ± 0.52
Cu		10.3 ± 0.8	8.55 ± 0.7	20.1 ± 1.6	23.6 ± 1.7
Co		<0.20	<0.20	1.33 ± 0.11	1.27 ± 0.10
Ni		2.76 ± 0.23	3.11 ± 0.25	3.82 ± 0.28	4.04 ± 0.31
Pb		5.84 ± 0.45	5.11 ± 0.40	7.80 ± 0.63	8.43 ± 0.71
Zn		23.5 ± 1.53	21.4 ± 1.46	36.5 ± 2.11	34.6 ± 2.23
Exc. Na^+^	mEq 100 g^−1^	1.94 ± 0.25	2.05 ± 0.18	4.12 ± 0.33	4.20 ± 0.35
Exc. Mg^2+^		4.80 ± 0.31	4.75 ± 0.32	1.48 ± 0.15	2.06 ± 0.18
Exc. Ca^2+^		56.8 ± 3.5	60.0 ± 3.1	43.3 ± 2.6	44.3 ± 2.7
Exc. K^+^		17.7 ± 1.2	18.1 ± 1.3	18.5 ± 1.2	18.1 ± 1.0
CEC		81.3 ± 3.7	84.9 ± 3.4	67.5 ± 2.9	68.6 ± 2.9

**Table 4 materials-16-01914-t004:** Filtration time and the main characteristics of unfiltered and filtered beer samples from laboratory tests.

Experiment	Filtration Time (min)	pH	Beer Color (EBC)	Beer Turbidity (EBC)	Beer Taste and Flavor
Unfiltered beer	-	4.20	14.1	10.33	normal
E1	17	4.31	12.3	1.51	normal
E2	16	4.40	11.2	1.25	normal
E3	18	4.34	9.37	1.12	normal
E4	22	4.40	9.29	0.93	normal
E5	24	4.40	9.32	0.95	normal
E6	20	4.35	9.44	0.83	normal
E7	26	4.36	9.41	0.80	normal
E8	25	4.36	8.64	0.67	normal
E9	27	4.38	6.78	0.74	normal
E10	21	4.38	8.95	1.00	normal
E11	23	4.40	8.94	0.94	normal
E12	20	4.36	8.56	1.11	normal
E13	22	4.28	8.32	1.05	normal
E14	28	4.36	7.55	0.58	normal
E15	28	4.30	6.05	0.49	normal
E16	20	4.46	8.93	0.61	normal
E17	23	4.30	8.62	0.55	normal
E18	19	4.36	8.32	0.90	normal
E19	23	4.31	8.16	0.75	normal
E20	26	4.33	6.95	0.44	normal
E21	27	4.38	6.51	0.37	normal

**Table 5 materials-16-01914-t005:** Contents of elements (Na, K, Ca, Mg, and Fe) in unfiltered and filtered beer samples measured by ICP-OES.

Experiment	Na	K	Ca	Mg	Fe
	mg L^−1^				
E0 (unfiltered beer)	13.1	459	21.0	69.4	1.33
E1	18.2	436	28.2	65.0	1.13
E2	12.9	399	18.9	61.9	0.88
E3	11.9	416	23.1	67.1	1.16
E4	15.9	513	22.6	77.7	1.46
E5	16.4	520	23.5	78.7	1.53
E6	12.5	523	26.6	80.4	0.89
E7	14.5	546	30.2	70.1	1.19
E8	11.6	580	53.8	70.9	0.81
E9	20.5	615	69.1	78.3	1.39
E10	18.0	569	53.4	72.9	0.81
E11	19.3	556	54.6	83.6	0.89
E12	15.9	540	41.1	79.0	0.82
E13	18.9	554	50.2	75.4	0.80
E14	13.0	518	43.6	79.4	0.58
E15	12.0	521	35.8	64.1	0.47
E16	13.6	483	37.7	71.9	0.52
E17	17.1	453	49.8	66.1	0.74
E18	14.2	502	30.9	75.8	0.40
E19	11.9	512	29.8	77.4	1.08
E20	10.9	491	29.3	75.6	0.38
E21	15.1	506	28.9	76.0	0.55

**Table 6 materials-16-01914-t006:** Contents of trace elements (As, Cd, Co, Cr, Cu, Ni, Pb, and Zn) in unfiltered and filtered beer samples measured by GFAAS.

Experiment	As	Cd	Co	Cr	Cu	Ni	Pb	Zn
	µg L^−1^							
E0 (unfiltered beer)	<0.50	<0.26	<0.65	2.80	57.9	10.9	5.55	10.6
E1	1.35	<0.26	<0.65	26.6	50.9	6.87	3.26	40.3
E2	1.30	<0.26	<0.65	23.1	33.9	5.27	1.18	29.5
E3	1.09	<0.26	<0.65	20.4	40.5	8.77	4.23	34.8
E4	0.89	<0.26	<0.65	21.4	43.2	10.7	5.36	44.1
E5	0.77	<0.26	<0.65	22.0	44.6	11.2	6.02	40.6
E6	0.68	<0.26	<0.65	18.4	26.3	5.74	4.02	28.8
E7	1.02	<0.26	<0.65	16.5	26.0	6.02	5.11	29.4
E8	2.44	<0.26	<0.65	18.1	25.4	9.28	8.14	35.8
E9	1.65	<0.26	<0.65	12.3	22.3	3.74	3.00	29.2
E10	1.89	<0.26	<0.65	17.9	28.4	8.28	7.44	30.6
E11	2.04	<0.26	<0.65	15.2	20.5	7.23	6.63	27.4
E12	3.20	<0.26	<0.65	17.2	28.8	3.24	4.29	24.6
E13	2.02	<0.26	<0.65	15.6	26.4	3.11	5.23	30.5
E14	1.12	<0.26	<0.65	11.6	20.2	3.91	3.18	19.7
E15	3.49	<0.26	<0.65	13.0	23.9	4.01	6.13	38.5
E16	3.11	<0.26	<0.65	17.9	28.4	4.54	1.88	36.6
E17	4.05	<0.26	<0.65	20.8	22.6	3.48	1.44	28.8
E18	3.21	<0.26	<0.65	22.3	24.3	3.79	1.61	35.0
E19	1.89	<0.26	<0.65	13.5	21.4	6.05	3.23	23.7
E20	1.76	<0.26	<0.65	13.3	28.3	4.53	4.43	26.4
E21	1.45	<0.26	<0.65	11.2	16.5	3.79	5.51	17.7

## Data Availability

The relevant data from this research are available in the authors’ repositories.

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
