# Peer review of "Preparation, Characterization, and Performance of Natural Zeolites as Alternative Materials for Beer Filtration"

_materials, 2023, doi:10.3390/ma16051914_

Round 1
Reviewer 1 Report
This paper investigates the performance of natural zeolites as a beer filtration. Some comments should be considered before its publication.
1 I suggest the authors to rewrite the abstract with a focus on background, objectives, methodology, main findings and conclusion. Please add a sentence which shows the necessity of the study.
2 Please add a photo about the collected zeolitic tuff samples.
3 Please add a photo or schematic diagram of the lab-scale filtration system.
4 The conclusion part should be more refined to make the findings and contributions of the paper clearer. Furthermore, please note the difference between the conclusions and abstract.
Reviewer 2 Report
The title of the paper has catched my eye, the idea is rather interesting. I looked through the literature and saw that zeolites were extensively reported to be good filters for brewery.
1) For this, please indicate vividly the novelty of the reseach. Was it just aimed at just “two deposits from North West Romania” or something else. I have just googled and found at leas ten works for the use of zeolites for beers.
2) [SiO4]4- and [AlO4]5- tetrahedra are confusing. Please remove charges.
3) I think that in the Introduction, a brief survey of the use of zeolites for filtering is needed.
4) “The concentrations of major elements” – please provide errors. “1.8895 (Al2O3)” etc. – excessive accuracy? Please also provide errors for the numeric values where applicable.
5) By the way, I think that the comparison of zeolites from “North West Romania” with the other zeolite materials may be of interest for the practical applications.
6) A brief illustration is needed to point the advantages of the used zeolites over traditional filtering materials.
7) For the industrial uses of filtering materials, the regeneration problem is of a key importance. The discussion must touch this issue.
8) Please comment on the statistics in the organoleptic experiments (the taste and flavour).
Reviewer 3 Report
The article, "Preparation, characterization, and performance of natural zeolites as alternative materials for beer filtration," presents the possibility of using natural zeolites from two deposits from northwestern Romania as an aid to beer filtration by improving its properties. For this purpose, the physical and chemical properties of heat-treated zeolites were determined. The filtration properties of natural zeolites and their mixtures with commercial filtration aids were studied in laboratory experiments using unfiltered beer samples. Each of the presented parts of the article is not questionable. The whole is coherent, closely related and presents results of high scientific value. In summary, each part of the publication has been described in detail and each contributes a great deal of substantive value to the subject matter presented by the authors. The conclusions are consistent and closely related to the research topic. As a reviewer of this work, I make no comments on the substantive and experimental content. I believe that the authors have exhausted all the topics included in the reviewed work.
Round 2
Reviewer 2 Report
The authors provided thorough step-by-step comments on my concerns. Now, I recommend the paper for publication.